# New and sex-specific migraine susceptibility loci identified from a multiethnic genome-wide meta-analysis

Hélène Choquet [1✉], Jie Yin[1], Alice S. Jacobson[2], Brandon H. Horton[1], Thomas J. Hoffmann [3,4], Eric Jorgenson [1], Andrew L. Avins [1,4] & Alice R. Pressman [2,4✉]

Migraine is a common disabling primary headache disorder that is ranked as the most common neurological cause of disability worldwide. Women present with migraine much more frequently than men, but the reasons for this difference are unknown. Migraine heritability is estimated to up to 57%, yet much of the genetic risk remains unaccounted for, especially in non-European ancestry populations. To elucidate the etiology of this common disorder, we conduct a multiethnic genome-wide association meta-analysis of migraine, combining results from the GERA and UK Biobank cohorts, followed by a European-ancestry meta-analysis using public summary statistics. We report 79 loci associated with migraine, of which 45 were novel. Sex-stratified analyses identify three additional novel loci (*CPS1*, *PBRM1*, and *SLC25A21*) specific to women. This large multiethnic migraine study provides important information that may substantially improve our understanding of the etiology of migraine susceptibility.

[1] Division of Research, Kaiser Permanente Northern California (KPNC), Oakland, CA, USA. [2] Sutter Health, Walnut Creek, CA, USA. [3] Institute for Human Genetics, University of California, San Francisco (UCSF), San Francisco, CA, USA. [4] Department of Epidemiology and Biostatistics, University of California, San Francisco (UCSF), San Francisco, CA, USA. ✉email: Helene.Choquet@kp.org; PressmAR@sutterhealth.org

Migraine is a common disabling disorder that can be accompanied by a wide range of symptoms of varying intensity, including headache pain that is often one-sided, and accompanied by nausea, sound and light sensitivity, and disturbed vision[1–3]. Epidemiological studies have repeatedly found that women have a substantially higher prevalence of migraine compared to men; however, the reasons for this difference are poorly understood[1,4–8].

Migraine has a moderate genetic component, with heritability estimates ranging between 0.10 and 0.57[9–16], depending on the type of estimate (i.e., based on twins studies vs. single-nucleotide polymorphism (SNP)-based heritability). In the past decade, genome-wide association studies (GWASs) have reported 42 genetic loci associated with migraine at genome-wide significance[17–22]. The most recent large genetic study of migraine was conducted by the International Headache Genetics Consortium (IHGC), by combining 22 European ancestry GWASs in a meta-analysis, they identified 38 loci, including 28 novel loci awaiting independent replication[21]. To our knowledge, no studies have yet conducted a genetic analysis of migraine in large and ethnically diverse cohorts. Therefore, there is a clear need for research to illuminate the genetic underpinnings of migraine.

Here, we present a large and ethnically diverse human genetic study of migraine, including, for the first time to our knowledge, East Asian, African American, and Hispanic/Latino adult individuals. Our study utilizes data from 554,569 individuals (28,852 migraine cases) from the Genetic Epidemiology Research in Adult Health and Aging (GERA) cohort and the UK Biobank (UKB) cohort (Table 1). We also used GWAS summary statistics data from the study of Gormley et al.[21], consisting of 375,752 participants (including 59,674 migraine cases) from the IHGC, which were publicly accessible. We conduct multiethnic GWAS meta-analyses, identifying several novel loci, including sex-specific ones. The associated loci provide potential causal variants, candidate genes, and relevant pathways underlying migraine susceptibility.

## Results

**Multiethnic meta-analysis of GERA and UKB**. We first undertook GWAS analysis of migraine in the GERA cohort and UKB cohort, stratified by ethnic group, followed by a meta-analysis combining results from GERA and UKB. This combined meta-analysis identified 22 loci associated with migraine ($P < 5 \times 10^{-8}$), of which 10 were novel (Fig. 1, Table 2, Supplementary Fig. 1, and Supplementary Data 1). The effect estimates of the ten lead SNPs at novel loci were consistent across the two studies (Supplementary Fig. 2), and no significant heterogeneity was observed between them (Table 2).

**Replication in the IHGC data**. We then tested the ten lead SNPs representing each of the ten novel loci for replication in the most recent large genetic study of migraine conducted by the IHGC[21]. However, as the GWAS summary statistics data, publicly accessible, reported only the SNPs from the Gormley et al. study with a $P$ value of less than $1.0 \times 10^{-5}$, some of the strongest SNPs reported by Gormley et al. were different than ours. Six loci, including *TMEM51*, *MIR4791-EFHB*, *LINC00472-RIMS1*, *FXN*, *GATA3-SFTA1P*, and *LINC00310-KCNE2*, replicated at Bonferroni significance ($P < 0.05/10$ novel loci $= 5.0 \times 10^{-3}$) (Supplementary Data 2). Our lead SNPs within the remaining four novel loci (i.e., *SLC45A1/RERE*, *MRGPRE-ZNF195*, *CALCB*, and *B3GNTL1-METRNL*) were not reported in the publicly accessible GWAS summary statistic from the Gormley et al. study; however, those may replicate at a Bonferroni significance threshold in IHGC ($1.0 \times 10^{-5} \leq P < 5.0 \times 10^{-3}$) but were not publicly accessible.

**Replication of previous migraine GWAS results**. We also investigated the lead SNPs within 38 loci associated with migraine at a genome-wide significance level from the most recent and exhaustive GWAS of migraine conducted to date[21] (Supplementary Data 3). Ten lead SNPs of the 36 available replicated at a genome-wide level of significance in our combined (GERA + UKB) multi-ethnic meta-analysis (including rs10218452 at *PRDM16*, rs2078371 near *TSPAN2/NGF*, rs1925950 at *MEF2D*, rs10166942 at *TRPM8/HJURP*, rs9349379 at *PHACTR1*, rs28455731 near *GJA1*, rs186166891 at *C7orf10*, rs6478241 at *ASTN2*, rs1024905 near *FGF6*, and rs11172113 at *LRP1*) (Supplementary Data 3). Furthermore, 14 additional SNPs replicated at Bonferroni significance ($P < 0.05/36 = 1.39 \times 10^{-3}$), and 4 showed nominal evidence of association ($P < 0.05$). In contrast, eight SNPs (including rs140002913 near *NOTCH4*, rs10155855 near *DOCK4/IMMP2L*, rs2506142 at *NRP1*, rs561561 at *IGSF9B*, rs75213074 near *WSCD1/NLRP1*, rs17857135 at *RNF213*, rs144017103 near *CCM2L/HCK*, and rs12845494 near *MED14/USP9X*) were not validated in the current combined (GERA + UKB) multiethnic meta-analysis ($P > 0.05$).

**Ethnic-specific and conditional analyses**. For ethnic groups represented in each cohort, we conducted ethnic-specific meta-analyses of each group. In the European ancestry (GERA non-Hispanic whites + UKB Europeans + IHGC Europeans only; 85,726 migraine cases and 803,292 controls) meta-analysis, we identified 73 loci, of which 35 were additional novel (Supplementary Data 4). To identify independent signals within the 73 genomic regions identified in the European-specific meta-analysis, we performed a multi-SNP-based conditional and joint association analysis (COJO)[23], which revealed two additional

## Table 1 Characteristics of the migraine cases and controls from GERA and UKB cohorts.

| | | GERA | | UKB | |
|---|---|---|---|---|---|
| | | **Controls** | **Cases** | **Controls** | **Cases** |
| Total, N (proportion that are cases) | | 60,282 | 11,320 (15.8%) | 465,435 | 17,532 (3.6%) |
| Age, mean ± SD | | 63.6 ± 13.5 | 58.9 ± 13.5 | 57.12 ± 8.10 | 55.80 ± 7.87 |
| Sex, N (proportion that are cases) | Female | 31,578 | 8982 (22.1%) | 248,184 | 13,518 (5.1%) |
| | Male | 28,704 | 2338 (7.5%) | 217,251 | 4014 (1.8%) |
| Ethnicity, N (proportion that are cases) | Non-Hispanic white/European | 49,036 | 9343 (16.0%) | 438,178 | 16,709 (3.6%) |
| | Hispanic/Latino | 4695 | 1142 (19.6%) | NA | NA |
| | East Asian | 4804 | 539 (10.1%) | 1815 | 30 (1.6%) |
| | African American/African | 1747 | 296 (14.5%) | 8357 | 208 (2.4%) |
| | South Asian | NA | NA | 9480 | 278 (2.8%) |
| | Mixed | NA | NA | 7605 | 307 (3.8%) |

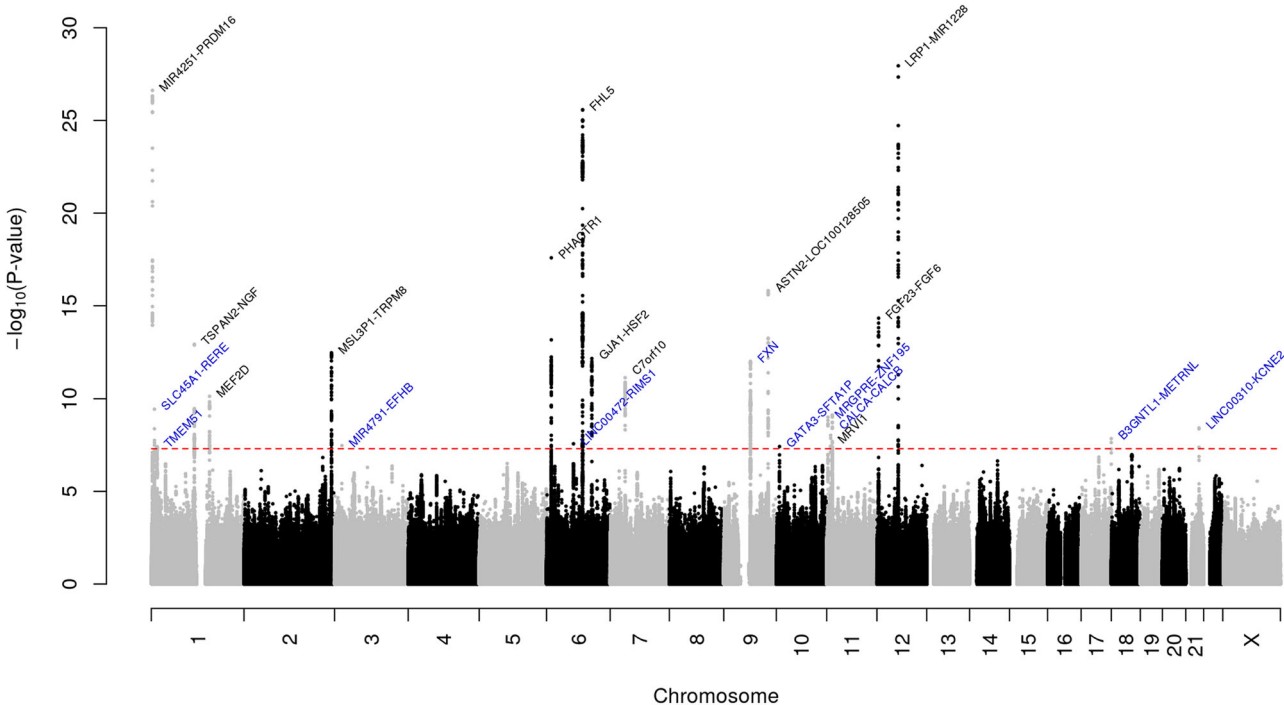

**Fig. 1 Manhattan plot of the multiethnic combined (GERA + UKB) GWAS meta-analysis of migraine.** The y-axis represents the −log$_{10}$(P value); all P values derived from logistic regression model are two-sided. The red dotted line represents the threshold of $P = 5 \times 10^{-8}$, which is the commonly accepted threshold of adjustments for multiple comparisons in GWAS. Locus names in blue are for the novel loci and the ones in dark are for the previously reported ones.

| Table 2 Migraine loci identified in the combined (GERA + UKB) GWAS multiethnic meta-analysis. | | | | | | | | |
|---|---|---|---|---|---|---|---|---|
| **SNP** | **Chr** | **Pos** | **Locus** | **EA /OA** | **OR (SE)** | **P** | **DE (GERA-UKB)** | **I² (Q)** |
| rs1393064 | 1 | 3099138 | *PRDM16* | C/G | 1.15 (0.013) | $2.38 \times 10^{-27}$ | ++ | 83.41 (0.014) |
| **rs10399665** | **1** | **8407287** | ***SLC45A1-RERE*** | **T/C** | **0.95 (0.009)** | **$3.67 \times 10^{-10}$** | −− | **0 (0.32)** |
| **rs10927732** | **1** | **15543202** | ***TMEM51*** | **C/A** | **1.07 (0.012)** | **$3.89 \times 10^{-8}$** | ++ | **0 (0.54)** |
| rs12134493 | 1 | 115677946 | *TSPAN2-NGF* | A/C | 1.10 (0.013) | $1.20 \times 10^{-13}$ | ++ | 88.35 (0.0034) |
| rs2274319 | 1 | 156450873 | *MEF2D* | C/T | 0.94 (0.0096) | $7.40 \times 10^{-11}$ | −− | 40.65 (0.19) |
| rs4663983 | 2 | 234815005 | *MSL3P1-TRPM8* | G/A | 0.92 (0.011) | $3.44 \times 10^{-13}$ | −− | 56.04 (0.13) |
| **rs13087932** | **3** | **19740933** | ***MIR4791-EFHB*** | **T/G** | **0.92 (0.016)** | **$3.38 \times 10^{-8}$** | −− | **0 (0.61)** |
| rs9349379 | 6 | 12903957 | *PHACTR1* | G/A | 0.92 (0.0092) | $2.59 \times 10^{-18}$ | −− | 93.14 ($1.0 \times 10^{-4}$) |
| **rs829470** | **6** | **72437128** | ***LINC00472-RIMS1*** | **C/T** | **1.05 (0.009)** | **$2.71 \times 10^{-8}$** | ++ | **0 (0.72)** |
| rs9486715 | 6 | 97059769 | *FHL5* | C/A | 1.10 (0.0093) | $2.62 \times 10^{-26}$ | ++ | 78.36 (0.032) |
| rs35136812 | 6 | 121815335 | *GJA1-HSF2* | GA/G | 1.09 (0.012) | $7.05 \times 10^{-13}$ | ++ | 27.1 (0.24) |
| rs549745067 | 7 | 40417816 | *C7of10* | T/TCTC | 1.10 (0.014) | $7.32 \times 10^{-12}$ | ++ | 79.41 (0.028) |
| **rs7855905** | **9** | **71700035** | ***FXN*** | **G/C** | **0.94 (0.0089)** | **$9.95 \times 10^{-13}$** | −− | **0 (0.74)** |
| rs7858153 | 9 | 119245085 | *ASTN2-LOC100128505* | A/G | 1.09 (0.010) | $1.54 \times 10^{-16}$ | ++ | 90.15 (0.0014) |
| **rs10795669** | **10** | **8720639** | ***GATA3-SFTA1P*** | **T/G** | **0.95 (0.010)** | **$3.81 \times 10^{-8}$** | −− | **0 (0.44)** |
| **rs10833535** | **11** | **3259478** | ***MRGPRE-ZNF195*** | **A/G** | **1.06 (0.0089)** | **$9.88 \times 10^{-10}$** | ++ | **0 (0.40)** |
| rs4909945 | 11 | 10673739 | *MRVI1* | C/T | 1.06 (0.0098) | $9.42 \times 10^{-9}$ | ++ | 0 (0.88) |
| **rs11023404** | **11** | **15004340** | ***CALCB*** | **T/C** | **1.06 (0.0092)** | **$7.68 \times 10^{-10}$** | ++ | **0 (0.77)** |
| rs2160875 | 12 | 4527322 | *FGF23-FGF6* | T/C | 0.93 (0.009) | $4.70 \times 10^{-15}$ | −− | 18.49 (0.27) |
| rs11172113 | 12 | 57527283 | *LRP1-MIR1228* | C/T | 0.90 (0.0092) | $1.12 \times 10^{-28}$ | −− | 87.29 (0.005) |
| **rs11655891** | **17** | **81015295** | ***B3GNTL1-METRNL*** | **G/A** | **1.07 (0.011)** | **$1.43 \times 10^{-8}$** | ++ | **0 (0.84)** |
| **rs79545715** | **21** | **35589328** | ***LINC00310-KCNE2*** | **C/T** | **0.90 (0.017)** | **$3.80 \times 10^{-9}$** | −− | **0 (0.41)** |

Loci in bold are novel.
*SNP* single-nucleotide polymorphism, *Chr* chromosome, *Pos* position, *EA* effect allele, *OA* other allele, *SE* standard error, *DE* direction of effect, *GERA* Genetic Epidemiology Research on Adult Health and Aging, *UKB* UK Biobank, *I²* heterogeneity index (0–100%), *Q* P value for Cochrane's Q statistic.

independent SNPs within the known loci *TSPAN2-NGF* (rs2207237) and *ADAMTSL4-ECM1* (rs7524797). Conducting a GWAS meta-analysis of East Asian-specific cohorts (GERA + UKB East Asian ancestry individuals only; 569 migraine cases and

6619 controls) and a GWAS meta-analysis of African-specific cohorts (GERA + UKB African ancestry individuals only; 504 migraine cases and 10,104 controls) did not result in the identification of genome-wide significant findings. We may have been

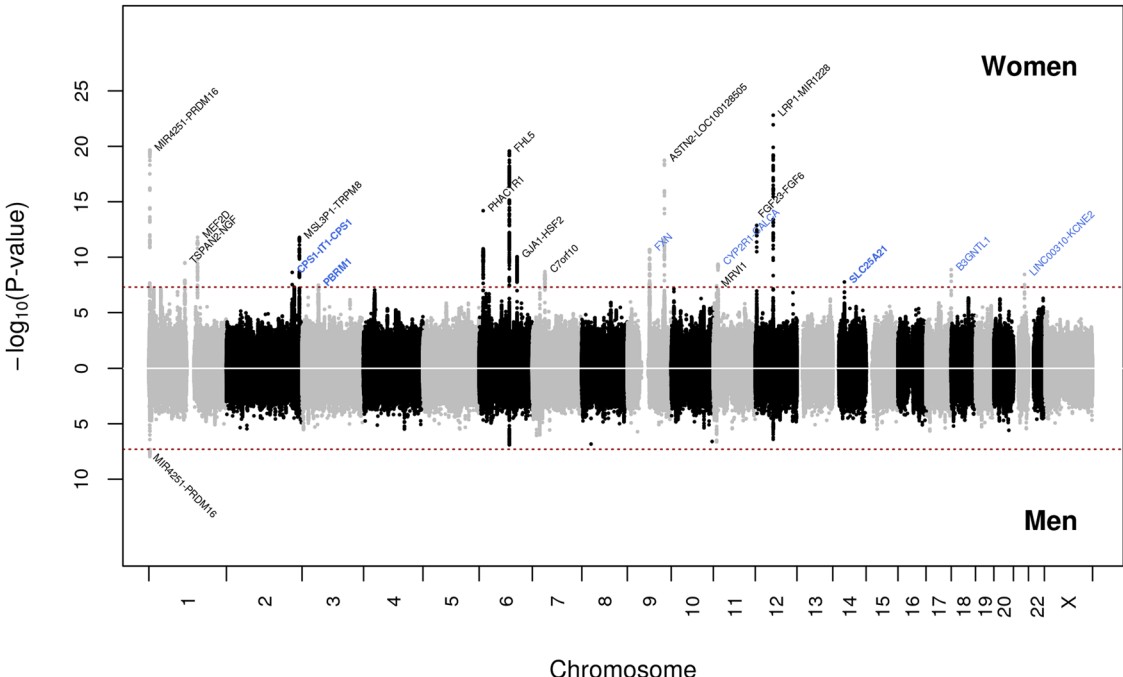

**Fig. 2 Chicago plot of the sex-stratified multiethnic GWAS meta-analyses of migraine.** Results from the meta-analysis combining women from GERA and UKB are presented on upper panel, while results from the meta-analysis combining men from GERA and UKB are presented on the lower panel. The *y*-axis represents the –log₁₀(*P* value); all *P* values derived from logistic regression model are two-sided. The red dotted line represents the threshold of *P* = 5 × 10⁻⁸, which is the commonly accepted threshold of adjustments for multiple comparisons in GWAS. Locus names in black are for those previously reported. Locus names in bold (i.e., *CPS1*, *PBRM1*, and *SLC25A21*) are for the additional novel loci specific to women (compared to the multiethnic meta-analysis (GERA + UKB)). Novel loci significantly associated (*P* < 5 × 10⁻⁸) with migraine in women are highlighted in blue.

underpowered to detect effects with statistical significance in those non-European ancestry meta-analyses.

**Sex-specific analyses identified additional loci**. Next, we conducted meta-analyses (GERA + UKB) for migraine by sex. The women-specific (GERA + UKB) meta-analysis (22,500 migraine cases and 279,762 controls) identified 19 loci ($P < 5.0 \times 10^{-8}$), of which three, *CPS1* on chromosome 2, *PBRM1* on chromosome 3, and *SLC25A21* on chromosome 14, were additional novel loci (Fig. 2 and Supplementary Data 5). *CPS1* rs1047891, *PBRM1* rs11718509, and *SLC25A21* rs10150336 were significantly associated with migraine in women ($P < 5.0 \times 10^{-8}$) but not in men ($P > 0.05$) (Supplementary Fig. 3a–c). Furthermore, among the loci also identified in the multiethnic meta-analysis (GERA + UKB), we observed that *ASTN2* locus was significantly differently associated in women and men (*ASTN2* rs7858153: OR = 1.11 and $P = 1.83 \times 10^{-19}$ in women; OR = 1.02, $P = 0.40$ in men; $Z = 3.59$, $P = 1.63 \times 10^{-4}$) (Supplementary Fig. 3d). Regional association plots illustrate the sex-specific association signals (Supplementary Fig. 3). For each of the four women-specific lead SNPs, we compared the effect allele frequencies (EAF) between cases and controls in women and men separately, as well as between women and men cases and women and men controls (Supplementary Data 6). We found that the EAF of each SNP were near identical when comparing women cases to men cases, or when comparing women controls to men controls. This shows that the differences in association signals in the women and men meta-analyses were not due to the difference in case and/or control EAF in the women and men samples. The men-specific (GERA + UKB) meta-analysis (6352 migraine cases and 245,955 controls) did not result in the identification of additional novel genome-wide significant findings (Fig. 2 and Supplementary

Data 7). To further evaluate the shared genetic basis of migraine between women and men, we compared the GWAS results from the two sex-specific meta-analyses by performing a linkage disequilibrium (LD) score regression (LDSC). We observed a high genetic correlation ($r_g$) between women and men for migraine ($r_g = 0.76$, $P = 2.39 \times 10^{-15}$).

**SNP prioritization and annotations**. To prioritize variants within the 22 loci identified in the combined multiethnic (GERA + UKB) meta-analysis and within the 73 loci identified in the combined (GERA + UKB + IHGC) European-specific meta-analysis, we applied a Bayesian approach (CAVIARBF)[24]. For each of the associated signals, we computed each variant's capacity to explain the identified signal within a 2 Mb window (±1.0 Mb with respect to the original top variant) and derived the smallest set of variants that included the causal variant with 95% probability (95% credible set). For the 22 loci identified in the combined multiethnic (GERA + UKB) meta-analysis, two sets included a unique variant (Supplementary Data 8). These include the previously reported intronic variant rs9349379 at *PHACTR1*[21], and the newly identified intergenic variant rs13087932 at *MIR4791-EFHB* with 100.0% and 97.2% posterior probability of being the causal variants, respectively, suggesting that these variants are more likely to be the true causal variants. For the 73 loci identified in the European-specific meta-analysis, four sets included a unique variant (Supplementary Data 9). In addition to rs9349379 at *PHACTR1*, we found that the intronic variants rs5763529 at *ASCC2* and rs11172113 at *LRP1*, and the intergenic variant rs28451064 at *LINC00310-KCNE2* were more likely to be the true causal variants with 100.0%, 99.9%, and 97.3% posterior probability, respectively.

**Gene-based association analysis and gene prioritization**. To identify additional genes associated with migraine at a gene level, we conducted a gene-based association analysis using the functional mapping and annotation of genetic associations (FUMA)[25] integrative tool using the combined multiethnic GWAS meta-analysis (GERA + UKB) results. FUMA implements Multi-marker Analysis of GenoMic Annotation (MAGMA)[26] gene-based analysis, which employs a multiple linear regression approach to properly incorporate LD between genetic variants and to detect multi-variant effects. As 19,933 genes were tested, the $P$ value adjusted for Bonferroni correction was set as $P < 2.51 \times 10^{-6}$ (0.05/19,933). We found significant associations with migraine for 47 genes, with the strongest association for *STAT6* ($P = 1.24 \times 10^{-23}$), followed by *UFL1* ($P = 7.26 \times 10^{-19}$), and *FHL5* ($P = 1.25 \times 10^{-18}$) (Supplementary Data 10). Out of the 47 genes, 9 were located outside the loci identified in the current study, including *PRKCE*, *RCHY1*, *THAP6*, *MAPK9*, *RP11-508N12.4*, *LRCH1*, *PNKP*, *AKT1S1*, and *TBC1D17*.

To prioritize genes within the 73 loci identified in the combined (GERA + UKB + IHGC) European-specific meta-analysis, we used the DEPICT[27] integrative tool. DEPICT gene prioritization analysis detected 15 genes, of which 9 were within novel migraine-associated loci, to prioritize after false-discovery rate (FDR) correction (Supplementary Data 11). These included: *LEPR* on chromosome 1, *TJP2* on chromosome 9, *AMBRA1* on chromosome 11, *HOXB2*, *HOXB3*, *HOXB6*, and *POLR2A* on chromosome 17, *TGFB1* on chromosome 19, and *JAG1* on chromosome 20.

**Biological pathway annotations and prioritization**. While DEPICT gene-set enrichment analysis using independent genome-wide significant genetic variants from the combined (GERA + UKB + IHGC) European-specific meta-analysis did not detect pathways to prioritize after FDR correction (Supplementary Data 12), FUMA[25] gene-set enrichment analysis based on the GWAS multiethnic meta-analysis (GERA + UKB) results highlighted many gene-sets involved in the flavonoid glucuronide biosynthetic process, the ascorbate and aldarate metabolic pathways, the response to endogenous stimulus or wounding, and the regulation of muscle contraction or system process (Supplementary Data 13).

FUMA tissue expression quantitative trait loci (eQTL) specificity analysis highlighted the sigmoid colon as the main tissue for which expression was affected by migraine-associated variants, followed by the esophagus muscularis and gastroesophageal junction ($P < 9.43 \times 10^{-4}$; Bonferroni significance after correcting for 53 GTEx tissues tested—Supplementary Fig. 4 and Supplementary Data 14). In contrast, DEPICT tissue-enrichment analysis using independent genome-wide significant genetic variants from the combined (GERA + UKB + IHGC) European-specific meta-analysis identified two tissues or cell type annotations to prioritize after FDR correction: the arteries (cardiovascular system), consistent with the previously reported Gormley et al. study[21], and the serous membrane (Supplementary Data 15).

**Genetic correlation between migraine and other phenotypes**. Genome-wide genetic correlations of migraine were calculated with a total of 772 complex traits and diseases by comparing allelic effects using a LDSC with the European-specific migraine meta-analysis (GERA + UKB) summary statistics (Methods). A total of 75 significant genetic correlations were observed ($P < 6.48 \times 10^{-5}$ which corresponds to 0.05/772 phenotypes tested; Supplementary Data 16). Among those 75 genetic correlations, 38 reached genome-wide level of significance (Supplementary Fig. 5). The strongest positive correlations were observed with

medications for pain relief, constipation, or heartburn; neck, shoulder or back pain; alcohol intake frequency; mood swings, anxiety, depression or neuroticism; and sleeplessness or insomnia. There was also a strong negative correlation with physical activity.

## Discussion

In this study, a multiethnic meta-analysis combining the GERA and UKB cohorts and a meta-analysis of migraine across European ancestry individuals identified 79 loci, of which 45 were novel. Our multiethnic meta-analysis also validated 78% of the migraine-associated loci identified in the most recent and exhaustive GWAS of migraine conducted to date[21]. Furthermore, our sex-stratified analyses identified three additional novel loci (*CPS1*, *PBRM1*, and *SLC25A21*) specific to women.

The identified loci give new insight and additional evidence about the genes and pathways/systems underlying migraine susceptibility. For instance, we identified a new region associated with migraine at *17q21* and our DEPICT gene analysis prioritized three members of the Antp homeobox family genes (i.e., *HOXB2*, *HOXB3*, and *HOXB6*) at this region that encode proteins with a homeobox DNA-binding domain. Those three genes have been involved in the early development[28–32] (i.e., hindbrain, nervous system, or epidermal development) and common variants in *HOXB3* have been shown to be associated with motion sickness, which is a condition that shares underlying genetic factors with migraine[33]. Our DEPICT gene analysis also prioritized *TGFB1* at the novel *19q13* migraine-associated locus. *TGFB1* encodes the transforming growth factor beta 1 protein that is a multi-functional proinflammatory cytokine that regulates cell proliferation, differentiation, and growth. Early works suggested that *TGFB1* could play a role in migraine susceptibility. Plasma level of *TGFB1* has been shown to increase in patients with migraine during headache-free periods compared to healthy subjects without headache[34]. Another study investigated the *TGFB1* genotype in pediatric migraine patients and reported significant differences between control and migraine patients[35]. Our DEPICT gene analysis also prioritized *JAG1* at the novel *20p12* migraine-associated locus. *JAG1* encodes that the jagged 1 protein is the ligand for the receptor notch 1, which is involved in signaling processes. *JAG1* plays a role in the formation of blood cellular components[36–38] and has been involved in the pathogenesis of patent foramen ovale, which is an atrial septal deformity associated with major causes of morbidity, including stroke and migraine[39,40]. Future investigations may provide insights into how these genes influence migraine susceptibility.

Our study also reported, for the first time to our knowledge, sex-specific loci associated with migraine susceptibility in women but not in men. Previous studies[18,41] evaluated the concordance of genetic risk for migraine between women and men but did not reveal any heterogeneity in the effect sizes of the genome-wide significant loci. Among the four sex-specific loci identified in the current study, two, *PBRM1* and *ASTN2*, have been involved in susceptibility to bipolar disorder and other psychiatric phenotypes. *PBRM1* encodes polybromo 1, which is a subunit of ATP-dependent chromatin-remodeling complexes. Common genetic polymorphisms at *PBRM1* (also named 3p21.1 locus) have been implicated in susceptibility to bipolar disorder, as well as major depression and schizophrenia[42–46]. *ASTN2* encodes astrotactin 2 that is expressed in the brain and is involved in neuronal migration. Deletions of *ASTN2* have been associated with schizophrenia and other psychiatric and neurodevelopmental disorders[47–49]. Similarly, our genetic correlation results indicate that migraine was significantly correlated with mood swings, anxiety, depression, or neuroticism, consistent with a previous genetic correlation analysis study[50]. Thus, these findings suggest

that migraine shares common variation risk with psychiatric disorders, such as schizophrenia, for which genome-wide significant loci involved in brain function have been reported[51,52].

Among the sex-specific loci associated with migraine susceptibility in women but not in men, less is known about the role of the identified *CPS1* and *SLC25A21* in regard to the biologic pathways underlying migraine. *CPS1* encodes a mitochondrial enzyme named carbamoyl-phosphate synthase 1 that catalyzes synthesis of carbamoyl phosphate from ammonia and bicarbonate. Mutations in this gene have been associated with metabolic deficiencies such as urea cycle disorder and hyperammonemia[53,54]. Individuals with a *CPS1* deficiency can present with a wide range of clinical manifestations, including headache, behavioral or psychiatric problems, learning disabilities, sleep disorder, periodic vomiting, seizures, coma, and even death[55–57]. *SLC25A21* (also known as *ODC1*) encodes the solute carrier family 25 member 21, which is essential for ammonium fixation and lysine biosynthesis[58]. While rare mutations in this gene cause a syndromic neurometabolic disorder associated with macrocephaly, developmental delay, alopecia, and dysmorphic features[59,60], common polymorphisms have been associated with smoking behaviors, and could contribute to Alzheimer's disease[61,62]. Thus, our findings will help to better understand the biological mechanisms underlying migraine susceptibility, particularly in women, even if follow-up experimental studies are needed to validate the role of the identified genes in migraine susceptibility.

Our results also illustrate the association between disorders of the large bowel and migraine susceptibility, as our eQTL analysis highlighted the sigmoid colon tissue for which expression was affected by migraine-associated variants. Similarly, our multiethnic meta-analysis results identified variants in *CALCB* associated with migraine susceptibility. *CALCB* encodes the calcitonin-related polypeptide beta (CGRP), which has been shown to contribute to migraine[63–65]. Several monoclonal antibodies targeting CGRP or its receptor have been proven to be effective therapeutics for the preventive treatment of migraine[66,67] and have been recently approved by the U.S. Food and Drug Administration[68,69]. In parallel, common genetic polymorphisms at *CALCB* have been reported to contribute to diverticular disease that is a common of intestinal neuromuscular function[70]. Of note, several epidemiologic studies reported that gastrointestinal disorders such as the "irritable bowel syndrome" are the most commonly reported comorbidities associated with migraine[71,72]. Our results suggest that the relationship between the intestine and migraine, also referred to "gut-brain axis," could have some genetic origin.

Our genetic correlation results identified the strongest positive correlations between migraine and neck, shoulder or back pain, suggesting shared genetic factors underlying those conditions. This result is consistent with early work demonstrating that migraine patients have higher pain responses in the splenius and temporalis muscles after a cognitive stress test compared to controls, and their muscle pain responses are regional (i.e., more pain in the neck and trapezius)[73]. Thus, our results support the concept that sensitization of pain pathways and the muscular system seem to be important in migraine susceptibility.

We recognize several potential limitations of our study. First, it is important to note phenotypic differences for migraine between the two study cohorts. Although migraine cases in GERA were identified in the Kaiser Permanente Northern California (KPNC) electronic health record system using our previously described and validated migraine probability algorithm (MPA)[74], which is based on migraine-specific prescriptions and International Classification of Disease, Ninth (ICD9) or Tenth Revision (ICD10) diagnosis codes, most of the migraine cases in UKB were based on

self-reported data. This may lead to phenotype misclassification that may have affected, for instance, the high positive genetic correlation between migraine and neck, shoulder, or back pain. However, our meta-analysis combining GERA and UKB results showed consistency of the SNPs effect estimates between cohorts. Furthermore, the previously reported associations[21] were well validated in our multiethnic meta-analysis and our novel loci were well replicated in the most recent large genetic study of migraine conducted by the IHGC[21]. Second, because of the limited sample of men cases (compared to women), we may have been underpowered to detect effects with statistical significance in the men-specific analysis.

In conclusion, our results identified additional loci that contribute to migraine susceptibility and represent potential candidates for the development of new therapeutic targets for this common neurological cause of disability. Although these findings will help to better understand the etiology of migraine susceptibility, additional genetic studies are needed to validate these associations in more large cohorts.

## Methods

**GERA**. The GERA cohort contains genome-wide genotype, clinical, and demographic data of over 110,000 adult members from mainly four ethnic groups (non-Hispanic white, Hispanic/Latino, East Asian, and African American) of the KPNC integrated healthcare system[75,76]. The Institutional Review Board of the Kaiser Foundation Research Institute approved all study procedures. Patients with migraine were identified in the KPNC electronic health record system using the previously validated MPA[74] (Supplementary Data 17), which is based on migraine-specific prescriptions and ICD9 diagnosis code: 346.XX and ICD10: G43.XX. We defined migraine cases as patients with a score >10 on the MPA (any evidence of migraine). After excluding individuals with any evidence of headache without a migraine diagnosis, as well as individuals with a MPA score = 10, our control group included all the non-cases. In total, 11,320 migraine cases and 60,282 controls from GERA were included in this study.

Protocols for participant genotyping, data collection and quality control (QC) have been described in detail[76]. Briefly, GERA participants' DNA samples were extracted from Oragene kits (DNA Genotek Inc., Ottawa, ON, Canada) at KPNC and genotyped at the Genomics Core Facility of the University of California, San Francisco. DNA samples were genotyped at over 665,000 genetic markers on four ethnic-specific Affymetrix Axiom arrays (Affymetrix, Santa Clara, CA, USA) optimized for European, Latino, East Asian, and African American individuals[77,78]. Genotype QC procedures and imputation were conducted on an array-wise basis[76], after an updated genotyping algorithm with an advanced normalization step specifically for SNPs in batches not recommended or flagged by the outlier plate detector than has previously been done. Subsequently, variants were excluded if: >3 clusters were identified; the number of batches was <38/42 (EUR array), <3/5 (AFR), <3/6 (EAS), or <7/9 (LAT); and the ratio of expected allele frequency variance across packages was <100 (EUR), <50 (AFR), <100 (EAS), <200 (LAT). On the EUR array, variants were additionally excluded if heterozygosity >0.52 or <0.02, and if an association test between Reagent kit v1.0 and v2.0 had $P < 10^{-4}$. Imputation was done by array, and we additionally removed variants with call rates <90%. Genotypes were then pre-phased with Eagle[79] v2.3.2, and then imputed with Minimac3[80] v2.0.1, using two reference panels. Variants were preferred if present in the EGA release of the Haplotype Reference Consortium (HRC; $n = 27,165$) reference panel[81], and from the 1000 Genomes Project Phase III release if not ($n = 2504$; e.g., indels)[82].

**UK Biobank**. The UKB is a large prospective study following the health of approximately 500,000 participants from five ethnic groups (European, East Asian, South Asian, African British, and mixed ancestries) resident in the UK aged between 40 and 69 years at the baseline recruitment visit[83,84]. Demographic information and medical history were ascertained through touch-screen questionnaires. Participants also underwent a wide range of physical and cognitive assessments, including blood sampling. Migraine cases ($N = 17,532$) were defined as participants with a self-reported migraine (data field 20002 code 1265) and/or a diagnosis code for migraine (ICD10: G43). After excluding participants who self-reported headaches (data field 20002, code 1436) and/or who had a diagnosis code for headaches (ICD10: G44), the control group included 465,435 participants who were not cases. Phenotyping, genotyping, and imputation were carried out by members of the UKB team. Imputation to the Haplotype Reference Consortium reference panel has been described (www.ukbiobank.ac.uk). Following QC, over 10 million variants in 487,622 individuals were tested for migraine adjusting for age, sex, and genetic ancestry principal components (PCs). The analyses presented in this paper were carried out under UKB Resource project #14105.

**The International Headache Genetics Consortium (IHGC)**. GWAS summary statistics data (SNPs with $P < 1.0 \times 10^{-5}$) from the study of Gormley et al.[21], consisting of 375,000 participants from the IHGC, were publicly accessible (http://eagle-i.itmat.upenn.edu/i/00000155-e1db-73aa-c956-e86e80000000).

**GWAS and adjustment in GERA**. We first analyzed each ethnic group (non-Hispanic white, Hispanic/Latino, East Asian, and African American) separately. We ran a logistic regression of migraine and each SNP using PLINK[85] v1.9 (www.cog-genomics.org/plink/1.9/) adjusting for age, sex, and ancestry PCs, which were previously[75] assessed within each ethnic group using Eigenstrat[86] v4.2. We included as covariates the top ten ancestry PCs for the non-Hispanic whites, whereas we included the top six ancestry PCs for the three other ethnic groups. To adjust for genetic ancestry, we also included the percentage of Ashkenazi (ASHK) ancestry as a covariate for the non-Hispanic white sample analyses[75]. For comparison, the GWAS analyses were also conducted using a new approach accounting for relatedness that fits a whole genome regression model, implemented in REGENIEv2.0.2[87] (https://rgcgithub.github.io/regenie/).

**GWAS meta-analyses**. First, a meta-analysis of migraine was conducted in GERA to combine the results of the 4 ethnic groups using the R[88] (https://www.R-project.org) package "meta." Similarly, a meta-analysis was conducted in UKB to combine the results of the five groups. The meta-analysis GWAS results generated using REGENIE were similar compared to the results generated using PLINK (Supplementary Data 18 and Supplementary Fig. 6). Three ethnic-specific meta-analyses were also performed: (1) combining European-specific samples (i.e., GERA non-Hispanic whites + UKB Europeans + IHGC Europeans); (2) combining East Asian-specific samples (i.e., GERA and UKB East Asians); and (3) combining African-specific samples (i.e., GERA African Americans and UKB Africans). A meta-analysis was then conducted to combine the results from GERA and UKB. The overall (GERA + UKB) meta-analysis results generated using REGENIE were similar compared to the results generated using PLINK (Supplementary Fig. 6). Two sex-specific meta-analyses were also performed: (1) combining women from GERA and UKB; and (2) combining men from GERA and UKB. Fixed-effects summary estimates were calculated for an additive model. We also estimated heterogeneity index, $I^2$ (0–100%) and $P$ value for Cochrane's Q statistic among different groups, and studies. For each locus, we defined the top SNP as the most significant variant within a 2 Mb window. Novel loci were defined as those that were located over 1 Mb apart from any previously reported locus[17,18,20,21].

**Conditional and joint (COJO) analysis**. A multi-SNP-based COJO[23] was performed on the combined European-specific (GERA non-Hispanic whites + UKB Europeans + IHGC Europeans) meta-analysis results to potentially identify independent signals within the 73 identified genomic regions. To calculate LD patterns, we used 10,000 randomly selected samples from GERA non-Hispanic white ethnic group as a reference panel. A $P$ value less than $5.0 \times 10^{-8}$ was considered as the significance threshold for this COJO analysis.

**Variants prioritization**. To prioritize variants within the identified genomic regions for follow-up functional evaluation, a Bayesian approach (CAVIARBF)[24] was used, which is available publicly at https://bitbucket.org/Wenan/caviarbf. Each variant's capacity to explain the identified signal within a 2 Mb window (±1.0 Mb with respect to the original top variant) was computed for each identified genomic region. Then the smallest set of variants that included the causal variant with 95% probability (95% credible set) was derived.

**FUMA for gene-based association analysis and gene-set analysis**. To prioritize genes and biological pathways, and highlight gene-set and tissue/cell enrichments within the 22 migraine-associated loci identified in the combined multiethnic (GERA + UKB) meta-analysis, we used the FUMA[25] integrative tool. FUMA uses input GWAS summary statistics to compute gene-based $P$ values (gene analysis) and gene-set $P$ value (gene-set analysis) using the MAGMA[26] tool, which employs multiple regression to obtain gene-based and gene-set $P$ values. GWAS summary statistics from the combined multiethnic (GERA + UKB) meta-analysis served as input for the "SNP2GENE" function. We used 10,000 random samples from GERA non-Hispanic whites as a reference panel. This "SNP2GENE" function provides extensive functional annotation for all SNPs in genomic areas identified and prioritizes gene-set and genes (using the MAGMA gene-set association analysis and the MAGMA gene analysis, respectively) for the next FUMA function, which is "GENE2FUNC." Then, this "GENE2FUNC" function annotates the prioritized genes in biological context. For gene analysis, the gene-based $P$ value is computed for protein-coding genes by mapping SNPs to genes if SNPs are located within the genes. For gene-set analysis, the gene-set $P$ value is computed using the gene-based $P$ value for 4728 curated gene-sets (including canonical pathways) and 6166 GO terms obtained from MsigDB v5.2. For both analyses, the default MAGMA setting (SNP-wise model for gene analysis and competitive model for gene-set analysis) is used, and the Bonferroni correction (gene) or FDR (gene-set) was used to correct for multiple testing. For the MAGMA gene-based association analysis conducted

on the combined (GERA + UKB) meta-analysis results, the $P$ value adjusted for Bonferroni correction was set as $P < 2.51 \times 10^{-6}$ (0.05/19,933 genes tested).

**FUMA tissue eQTL specificity**. To highlight and visualize tissue eQTL enrichments within the 22 migraine-associated genomic regions identified in the combined multiethnic (GERA + UKB) meta-analysis, we used the FUMA[25] integrative tool. FUMA is an integrative web-based platform that accommodates eQTL, and provides tissue enrichment results for each of 53 tissue types based on the genotype-tissue expression (GTEx) v6 RNA-seq data[89].

**DEPICT**. To prioritize genes and biological pathways, and highlight gene-set and tissue/cell enrichments within the 73 migraine-associated loci identified in the combined (GERA + UKB + IHGC) European meta-analysis, we used the following integrative tool: DEPICT[27]. All independent genome-wide significant genetic variants ($P < 5.0 \times 10^{-8}$) served as input, and as the reference panel, we used 10,000 random samples from GERA non-Hispanic white ethnic group. Genes, gene-sets, and tissue/cell annotations that achieved a nominal significance level of 0.05 after FDR correction were subsequently prioritized.

**Genetic correlations**. To estimate the genetic correlation of migraine with more than 700 diseases/traits from different publicly available resources/consortia, we used the LD Hub web interface[90], which performs automated LDSC. In the LDSCs, we included only HapMap3 SNPs with MAF > 0.01. Genetic correlations were considered significant after Bonferroni adjustment for multiple testing ($P < 6.48 \times 10^{-5}$ which corresponds to 0.05/772 phenotypes tested).

**Reporting summary**. Further information on research design is available in the Nature Research Reporting Summary linked to this article.

## Data availability

The GERA genotype data are available upon application to the KP Research Bank (https://researchbank.kaiserpermanente.org/). The combined multiethnic (GERA + UKB) meta-analysis GWAS summary statistics are available from the NHGRI-EBI GWAS Catalog (https://www.ebi.ac.uk/gwas/downloads/summary-statistics), study accession number GCST90000016. GWAS summary statistics data (SNPs with $P < 1.0 \times 10^{-5}$) from the study of Gormley et al.[21] are publicly accessible (http://eagle-i.itmat.upenn.edu/i/00000155-e1db-73aa-c956-e86e80000000).

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

## Acknowledgements

We are grateful to the Kaiser Permanente Northern California members who have generously agreed to participate in the Kaiser Permanente Research Program on Genes, Environment, and Health. Support for participant enrollment, survey completion, and biospecimen collection for the RPGEH was provided by the Robert Wood Johnson Foundation, the Wayne and Gladys Valley Foundation, the Ellison Medical Foundation, and Kaiser Permanente Community Benefit Programs. Genotyping of the GERA cohort was funded by a grant from the National Institute on Aging, National Institute of Mental Health, and National Institute of Health Common Fund (RC2 AG036607). H.C. and E.J. were supported by the National Eye Institute (NEI) grant R01 EY027004, the National Institute of Diabetes and Digestive and Kidney Diseases (NIDDK) R01 DK116738, and by the National Cancer Institute (NCI) R01CA2416323. The Genetics and Comorbidity of Migraine study was funded by a grant from the National Institute for Neurological Disorders and Stroke (NINDS) R01NS080863.

## Author contributions

H.C., A.L.A., and A.R.P. contributed to study conception and design. T.J.H. and E.J. were involved in the genotyping and quality control of the GERA samples. T.J.H. performed the imputation analyses in the GERA cohort. A.S.J. and B.H.H. extracted phenotype and other data for the GERA subjects based on EHRs. J.Y. performed statistical analyses and in silico analyses. H.C., A.L.A., and A.R.P. interpreted the results of analyses. H.C. drafted the manuscript. H.C., T.J.H., E.J., A.L.A., and A.R.P. critically revised the manuscript for key intellectual content.

## Competing interests

H.C. is an Editorial Board Member for *Communications Biology*, but was not involved in the editorial review of, or the decision to publish this article. The authors declare no other competing interests.
