## [Peer Review File · Communications Biology]

Reviewers' comments:

Reviewer #1 (Remarks to the Author):

The authors report results from genome-wide association (GWA) analysis of a novel multi-ethnic cohort. Their results have potential to provide important new knowledge on the genetics of migraine; however, I note the following comments that need addressing.

The authors should review and discuss the paper by Chang et al (2018) which conducted a migraine GWAS in a sample of African-American cases and controls [PMID: 30266756]. The authors should also note and discuss the Anttila et al (2013) EU migraine GWAS which conducted a heterogeneity analysis between men and women at the 146 index SNPs (with a P-value < 1E-05) [PMID: 23793025]; and Nyholt et al (2015) which assessed the genetic overlap (SNP effect concordance) across female and migraine GWAS data [PMID: 25179292].

The manuscript currently emphasises results indicating sex-specific SNP risk loci; however, the data as presented lack the necessary details to unequivocally support such an interpretation. In particular, the authors need to provide details for the effect (A1?) and non-effect (A2?) allele frequencies in the cases and controls for the male- and female-specific GWAS. In particular, the authors should confirm and discuss whether the difference in association signals in the female and male GWAS are driven (due to) the difference in case and/or control allele frequencies in the male and female samples. Indeed, I would suggest the authors directly compare the female to male cases frequencies (and female to male control frequencies) to show that the difference in association signal is be driven by case frequencies

Re SNP (risk allele) effect, the authors should clarify which allele the effects (odds ratios [ORs]) refer to both in the text and tables (e.g., Table 2 lists alleles [e.g., "C/G"] but does not specify which allele the OR refers to.

Indeed, the authors need to provide details (definitions) for ALL the columns in their tables (including Suppl Tables).

In particular, in addition to the p-values and ORs provided in 'Supplementary Table 2. Look-up of the 41 previously reported migraine loci in the combined (GERA+UKB) multiethnic analysis results', the authors should clearly define/tabulate the ORs of the previously reported migraine along with the ORs in their GERA+UKB analysis to confirm the allelic effect are in the same direction as well as having a significant p-value (i.e., truly replicate).

Lastly, I generally found the Discussion slightly superficial as it did not sufficiently delve into the potential mechanisms (relevance) of the newly identified SNP (and associated gene) loci and pathways. The authors need to perform a more comprehensive literature search and interpretation of their GWAS findings to provide greater insight into the importance of their findings.

Reviewer #3 (Remarks to the Author):

In this study, Choquet et al. investigated the genetic architecture of migraine by performing a multi-ethnic genome-wide association study (GWAS) on migraine in the UKB and GERA, which was then meta-analysed with the previously performed IHGC GWAS. To gain more insight into possible reasons for the difference in migraine prevalence between men and women, they additionally performed sex-specific GWAS. This is an important study reporting the largest GWAS on migraine thus far, identifying an additional ~30 associated loci, which is of relevance for other researchers in the field. There are however some points that need addressing prior to publication, which will be covered in a point-by-point fashion below.

Major points:

1. Particularly the meta-analysis combined with the IHGC data is of importance for the field. Although I understand the approach of first meta-analysing UKB+GERA prior to meta-analysing

with the IHGC summary statistics, the follow-up analyses (from SNP prioritization onwards) are again performed in the first UKB+GERA set. Since the largest meta-analysis including IHGC has the most power, I think it would be better to perform all follow-up analyses in the complete set including IHGC. This could replace the current follow-up analyses, or could be performed in addition to the current follow-up analyses.

2. The authors aim to answer a very relevant question in the migraine field, namely whether genetic factors can partially explain differences in migraine prevalence between men and women. They performed sex-specific GWAS to answer this question, but there are additional analyses that could be performed to provide more answers to this question. For example, it might be interesting to check the genetic correlation between the two meta-analyses, or to investigate whether different pathways play a role in the two.

3. In the Methods section, the authors write they have included 487k UKB participants in the GWAS, which was performed in PLINK. However, there are many related individuals present in the UKB. These related individuals should either be excluded from the analysis if a normal logistic regression is performed, or software that takes into consideration the genetic relationships should be used – such as BOLT-LMM. Could the authors explain how they have dealt with the relatedness in this sample?

Minor points:

4. (Abstract) In line with the first point, my suggestion would be to mention the number of identified loci of the meta-analysis including IHGC, and the number of novel loci of this combined GWAS compared to Gormley et al.

5. (Introduction) '[...] and severe disruptions of the brain parenchyma'.

It is not generally accepted that the brain parenchyma is severely disrupted by migraine, the authors could consider to rephrase this sentence.

6. (Results – Multiethnic meta-analysis of GERA and UKB) Similar to the replication section, it would be interesting to replicate the UKB+GERA findings in the IHGC GWAS using a Bonferroni significance threshold.

7. (Results – Replication of previous migraine GWAS results) My suggestion would be to only to consider the meta-analysis performed by Gormley et al. since this is the most recent and most complete GWAS which includes the previously performed GWAS.

8. (Results – Ethnic-specific and conditional analyses) Please add the number of cases and controls for each ancestry-specific meta-analysis, as this is probably the most important reason why no additional loci have been identified in the non-European ancestry analyses.

9. (Results – Ethnic-specific and conditional analyses) Although no novel loci were identified in the non-European ancestry analysis, could the authors provide some information on whether the identified loci in the multi-ethnic analysis was still present in the ancestry-specific analyses?

10. (Results – SNP prioritization and annotations) Could the authors provide some more information on the two likely causal variants, e.g. are they exonic/intronic, what is their CADD score, etc.?

11. (Results – Gene-Based Association Analysis) Since gene-based analyses take a different approach and have a more lenient threshold, were there also genes identified outside the originally identified loci?

12. (Results – Genetic correlation between migraine and other phenotypes) Why was a P-value threshold of $<5 \times 10^{-8}$ used for this analysis?

13. (Discussion) Do the authors think the genetic correlation with neck, shoulder and back pain may in part be explained by a misclassification of migraine cases, with some being actually tension headaches rather than migraine?

Reviewers' comments:

Reviewer #1 (Remarks to the Author):

The authors report results from genome-wide association (GWA) analysis of a novel multi-ethnic cohort. Their results have potential to provide important new knowledge on the genetics of migraine; however, I note the following comments that need addressing.

We thank the reviewer for the positive feedback and the helpful comments.

The authors should review and discuss the paper by Chang et al (2018) which conducted a migraine GWAS in a sample of African-American cases and controls [PMID: 30266756]. The authors should also note and discuss the Anttila et al (2013) EU migraine GWAS which conducted a heterogeneity analysis between men and women at the 146 index SNPs (with a P-value < 1E-05) [PMID: 23793025]; and Nyholt et al (2015) which assessed the genetic overlap (SNP effect concordance) across female and migraine GWAS data [PMID: 25179292].

We have now cited the paper by Chang et al (2018) in the Introduction, as follows:

“In the past decade, genome-wide association studies (GWASs) have reported 42 genetic loci associated with migraine at genome-wide significance¹⁻⁶.”

We also specified in the Introduction that our study includes adult individuals (in contrast to the Chang et al (2018) study which conducted a migraine GWAS in African-American children):

“Here, we present a large and ethnically diverse human genetic study of migraine, including, for the first time to our knowledge, East Asian, African American and Hispanic/Latino adult individuals.”

We have also discussed the Anttila et al (2013) and Nyholt et al (2015) in the Discussion, as follows:

“Our study also reported, for the first time to our knowledge, sex-specific loci associated with migraine susceptibility in women but not in men. Previous studies^{2, 7} evaluated the concordance of genetic risk for migraine between women and men but did not reveal any heterogeneity in the effect sizes of the genome-wide significant loci.”

The manuscript currently emphasises results indicating sex-specific SNP risk loci; however, the data as presented lack the necessary details to unequivocally support such an interpretation. In particular, the authors need to provide details for the effect (A1?) and non-effect (A2?) allele frequencies in the cases and controls for the male- and female-specific GWAS. In particular, the authors should confirm and discuss whether the difference in association signals in the female and male GWAS are driven (due to) the difference in case and/or control allele frequencies in the male and female samples. Indeed, I would suggest the authors directly compare the female to male cases frequencies (and female to male control frequencies) to show that the difference in association signal is be driven by case frequencies

We confirm that ‘A1’ corresponds to the effect allele (EA) and ‘A2’ to the other allele. We now provide this information along with the effect allele frequencies (EAF) in the cases and controls for the male- and female-specific lead SNPs in the Supplementary Datas 5 and 7, as well as in the table below for the 4 lead SNPs that show sex differences in effect sizes and significance of association. For of the 4 women-specific lead SNPs, we compared the EAF between cases and controls in women and men separately:

Locus	SNP	EA	Women (N=302,262)			Men (N=252,307)		
			EAF Cases (N=22,500)	EAF Controls (N=279,762)	Chi-2 P-value	EAF Cases (N=6,352)	EAF Controls (N=245,955)	Chi-2 P-value
CPS1	rs1047891	A	0.32	0.31	0.0006	0.31	0.31	0.86
PBRM1	rs11718509	A	0.38	0.39	0.19	0.39	0.38	0.22
SLC25A21	rs10150336	T	0.51	0.52	0.0005	0.52	0.52	0.79
ASTN2	rs7858153	A	0.24	0.23	6.59x10 ⁻⁹	0.23	0.23	0.60

N, number of individuals

We also compared the EAF between women and men cases and women and men controls:

Locus	SNP	EA	Cases (N=28,852)			Controls (N=525,717)		
			EAF Women (N=22,500)	EAF Men (N=6,352)	Chi-2 P-value	EAF Women (N=279,762)	EAF Men (N=245,955)	Chi-2 P-value

CPS1	rs1047891	A	0.32	0.31	0.13	0.31	0.31	0.14
PBRM1	rs11718509	A	0.38	0.39	0.20	0.39	0.38	0.016
SLC25A21	rs10150336	T	0.51	0.52	0.21	0.52	0.52	0.31
ASTN2	rs7858153	A	0.24	0.23	0.020	0.23	0.23	0.98

We found that the EAF of each SNP were near identical across the 4 sub-groups (i.e. women and men cases, and women and men controls) and we added some text in the Results section to reflect this new evidence that the differences in association signals in the women and men meta-analyses were not due to the difference in case and/or control EAF in the women and men samples, as follows:

“Sex-specific analyses identified additional loci

(...)

For each of the 4 women-specific lead SNPs, we compared the effect allele frequencies (EAF) between cases and controls in women and men separately, as well as between women and men cases and women and men controls (Supplementary Data 6). We found that the EAF of each SNP were near identical across the 4 sub-groups (i.e. women and men cases, and women and men controls). This shows that the differences in association signals in the women and men meta-analyses were not due to the difference in case and/or control EAF in the women and men samples.”

Re SNP (risk allele) effect, the authors should clarify which allele the effects (odds ratios [ORs]) refer to both in the text and tables (e.g., Table 2 lists alleles [e.g., “C/G”] but does not specify which allele the OR refers to. Indeed, the authors need to provide details (definitions) for ALL the columns in their tables (including Suppl Tables).

We confirm that in all the tables, we reported the ORs corresponding to the ‘A1’ (effect allele). Further, for Table 2 and Supplementary Datas 1, 4, and 5, we now provide this information and other details (i.e. abbreviations) and we have renamed the columns ‘Effect /Other Allele’ instead of ‘A1/A2’.

In particular, in addition to the p-values and ORs provided in ‘Supplementary Data 2. Look-up of the 41 previously reported migraine loci in the combined (GERA+UKB) multiethnic analysis results’, the authors should clearly define/tabulate the ORs of the previously reported migraine along with the ORs in their GERA+UKB analysis to confirm the allelic effect are in the same direction as well as having a significant p-value (i.e., truly replicate).

We now provide the ORs of the previously reported migraine-associated lead SNPs (from the Gormley et al. study) along with the ORs from the combined (GERA+UKB) meta-analysis in the **Supplementary Data 3**. We have also added a column reporting the direction of effect to confirm that the allelic effects are in the same direction across the 2 studies.

Lastly, I generally found the Discussion slightly superficial as it did not sufficiently delve into the potential mechanisms (relevance) of the newly identified SNP (and associated gene) loci and pathways. The authors need to perform a more comprehensive literature search and interpretation of their GWAS findings to provide greater insight into the importance of their findings.

To gain biological insights from our GWAS of migraine and to provide some interpretation of our GWAS findings, we used DEPICT⁸, an integrative tool that employs predicted gene functions to systematically prioritize the most likely causal genes at associated loci, highlight enriched pathways and identify tissues/cell types where genes from associated loci are highly expressed. We conducted our DEPICT analyses using as input independent genome-wide significant genetic variants from the combined (GERA+UKB+IHGC) European-specific meta-analysis.

We now present those new results in the Supplementary Information as well as in the Results section, as follows:

“Gene-Based Association Analysis and Gene Prioritization

To prioritize genes within the 73 loci identified in the combined (GERA+UKB+IHGC) European-specific meta-analysis, we used the DEPICT⁸ integrative tool. DEPICT gene prioritization analysis detected 15 genes, of which 9 were within novel migraine-associated loci, to prioritize after false discovery rate (FDR) correction (Supplementary Data 11). These included: *LEPR* on chromosome 1, *TJP2* on chromosome 9, *AMBRA1* on chromosome 11, *HOXB2*, *HOXB3*, *HOXB6*, and *POLR2A* on chromosome 17, *TGFB1* on chromosome 19, and *JAG1* on chromosome 20.

Biological pathway annotations and prioritization

While DEPICT gene-set enrichment analysis using independent genome-wide significant genetic variants from the combined (GERA+UKB+IHGC) European-specific meta-analysis did not detect pathways to prioritize after FDR correction (Supplementary Data 12), (...) DEPICT tissue-enrichment analysis using independent genome-wide significant genetic variants from the combined (GERA+UKB+IHGC) European-specific meta-analysis identified 2 tissues or cell type annotations to prioritize after FDR correction: the arteries (cardiovascular system), consistent with the previously reported Gormley et al. study⁵, and the serous membrane (Supplementary Data 13)."

Those important DEPICT results gave us the opportunity to delve into the potential mechanisms (relevance) of the newly identified loci (and prioritized genes) and to improve the Discussion:

"The identified loci give new insight and additional evidence about the genes and pathways/systems underlying migraine susceptibility. For instance, we identified a new region associated with migraine at 17q21 and our DEPICT gene analysis prioritized 3 members of the Antp homeobox family genes (i.e. HOXB2, HOXB3, and HOXB6) at this region that encode proteins with a homeobox DNA-binding domain. Those 3 genes have been involved in the early development⁹⁻¹³ (i.e. hindbrain, nervous system, or epidermal development) and common variants in HOXB3 have been shown to be associated with motion sickness, which is a condition that shares underlying genetic factors with migraine¹⁴. Our DEPICT gene analysis also prioritized TGFB1 at the novel 19q13 migraine-associated locus. TGFB1 encodes the transforming growth factor beta 1 protein which is a multifunctional proinflammatory cytokine that regulates cell proliferation, differentiation and growth. Early works suggested that TGFB1 could play a role in migraine susceptibility. Plasma level of TGFB1 has been shown to increase in patients with migraine during headache-free periods compared to healthy subjects without headache¹⁵. Another study investigated the TGFB1 genotype in pediatric migraine patients and reported significant differences between control and migraine patients¹⁶. Our DEPICT gene analysis also prioritized JAG1 at the novel 20p12 migraine-associated locus. JAG1 encodes the jagged 1 protein is the ligand for the receptor notch 1, which is involved in signaling processes. JAG1 plays a role in the formation of blood cellular components¹⁷⁻¹⁹ and has been involved in the pathogenesis of patent foramen ovale, which is an atrial septal deformity associated with major causes of morbidity, including stroke and migraine^{20, 21}. Future investigations may provide insights into how these genes influence migraine susceptibility."

Further, after performing a more comprehensive literature search, we have expanded the Discussion, and for instance, we discussed the previous evidence for our identified locus CALCB (that encodes CGRP which is an authorized migraine preventive treatment), as follows:

"(...) our multiethnic meta-analysis results identified variants in CALCB associated with migraine susceptibility. CALCB encodes the calcitonin related polypeptide beta (CGRP), which has been shown to contribute to migraine²²⁻²⁴. Several monoclonal antibodies targeting CGRP or its receptor have been proven to be effective therapeutics for the preventive treatment of migraine^{25, 26} and have been recently approved by the U.S. Food and Drug Administration^{27, 28}."

Reviewer #3 (Remarks to the Author):

In this study, Choquet et al. investigated the genetic architecture of migraine by performing a multi-ethnic genome-wide association study (GWAS) on migraine in the UKB and GERA, which was then meta-analysed with the previously performed IHGC GWAS. To gain more insight into possible reasons for the difference in migraine prevalence between men and women, they additionally performed sex-specific GWAS. This is an important study reporting the largest GWAS on migraine thus far, identifying an additional ~30 associated loci, which is of relevance for other researchers in the field. There are however some points that need addressing prior to publication, which will be covered in a point-by-point fashion below.

We thank the reviewer for the positive feedback and constructive review.

Major points:

1. Particularly the meta-analysis combined with the IHGC data is of importance for the field. Although I understand the approach of first meta-analysing UKB+GERA prior to meta-analysing with the IHGC summary statistics, the follow-up analyses (from SNP prioritization onwards) are again performed in the first UKB+GERA set. Since the largest meta-analysis including IHGC has the most power, I think it would be better to perform all follow-up analyses in the complete set including IHGC. This could replace the current follow-up analyses, or could be performed in addition to the current follow-up analyses.

We agree with the reviewer that the largest meta-analysis including IHGC has the most power, and we have now conducted follow-up analyses (i.e. CAVIAR, DEPICT) based on the meta-analysis (UKB+GERA+IHGC) results and presented

those results in the Supplementary Information in addition to the current ones. We have also added some text in the Results section as follows:

“SNP prioritization and annotations

To prioritize variants within the 22 loci identified in the combined multiethnic (GERA+UKB) meta-analysis and within the 73 loci identified in the combined (GERA+UKB+IHGC) European-specific meta-analysis, we applied a Bayesian approach (CAVIARBF)²⁹. (...) For the 73 loci identified in the European-specific meta-analysis, four sets included a unique variant (**Supplementary Data 9**). In addition to rs9349379 at *PHACTR1*, we found that the intronic variants rs5763529 at *ASCC2* and rs11172113 at *LRP1*, and the intergenic variant rs28451064 at *LINC00310-KCNE2* were more likely to be the true causal variants with 100.0%, 99.9%, and 97.3% posterior probability, respectively.”

“Gene-Based Association Analysis and Gene Prioritization

(...)

To prioritize genes within the 73 loci identified in the combined (GERA+UKB+IHGC) European-specific meta-analysis, we used the DEPICT⁸ integrative tool. DEPICT gene prioritization analysis detected 15 genes, of which 9 were within novel migraine-associated loci, to prioritize after false discovery rate (FDR) correction (**Supplementary Data 10**). These included: *LEPR* on chromosome 1, *TJP2* on chromosome 9, *AMBRA1* on chromosome 11, *HOXB2*, *HOXB3*, *HOXB6*, and *POLR2A* on chromosome 17, *TGFB1* on chromosome 19, and *JAG1* on chromosome 20.”

“Biological pathway annotations and prioritization

While DEPICT gene-set enrichment analysis using independent genome-wide significant genetic variants from the combined (GERA+UKB+IHGC) European-specific meta-analysis did not detect pathways to prioritize after FDR correction (**Supplementary Data 12**), (...) DEPICT tissue-enrichment analysis using independent genome-wide significant genetic variants from the combined (GERA+UKB+IHGC) European-specific meta-analysis identified 2 tissues or cell type annotations to prioritize after FDR correction: the arteries (cardiovascular system) and the serous membrane (**Supplementary Data 15**).”

2. The authors aim to answer a very relevant question in the migraine field, namely whether genetic factors can partially explain differences in migraine prevalence between men and women. They performed sex-specific GWAS to answer this question, but there are additional analyses that could be performed to provide more answers to this question. For example, it might be interesting to check the genetic correlation between the two meta-analyses, or to investigate whether different pathways play a role in the two.

To further support our sex-specific GWAS findings, and as suggested by the reviewer, we looked at the genetic correlation between the two sex-specific meta-analyses by performing a LD score regression (LDSC). As expected, we found a high genetic correlation for migraine between women and men ($r_g=0.76$, $P=2.39 \times 10^{-15}$). We have added some text in the Results section as below:

“Sex-specific analyses identified additional loci

(...) To further evaluate the shared genetic basis of migraine between women and men, we compared the GWAS results from the two sex-specific meta-analyses by performing a LD score regression (LDSC). We observed a high genetic correlation (r_g) between women and men for migraine ($r_g=0.76$, $P=2.39 \times 10^{-15}$).”

To investigate whether different pathways play a role in migraine susceptibility according to sex, we conducted a gene-set enrichment analysis in women and men separately using the DEPICT integrative tool. The women-specific DEPICT gene-set enrichment analysis did not detect pathway to prioritize after false discovery rate (FDR) correction. As no genetic variants reached genome-wide level of significance in the men-specific meta-analysis (and those served as input to conduct the DEPICT analysis), we were not able to conduct the men-specific DEPICT gene-set enrichment analysis.

3. In the Methods section, the authors write they have included 487k UKB participants in the GWAS, which was performed in PLINK. However, there are many related individuals present in the UKB. These related individuals should either be excluded from the analysis if a normal logistic regression is performed, or software that takes into consideration the genetic relationships should be used – such as BOLT-LMM. Could the authors explain how they have dealt with the relatedness in this sample?

This is an excellent point raised by the reviewer. In addition to conduct the GWAS analyses for migraine using PLINK, we also performed the GWAS analyses using a new approach which accounts for relatedness, named REGENIE³⁰ (available at <https://rgcg.github.io/regenie/>). The REGENIE results were similar compared to the PLINK results and we have reported those results comparison in a Supplementary Data (Supplementary Data 18). We have also added some text in the Methods section as follows:

“For comparison, the GWAS analyses were also conducted using a new approach accounting for relatedness that fits a whole genome regression model, implemented in REGENIEv2.0.2³⁰ (<https://rgcg.github.io/regenie/>). The GWAS results generated using REGENIE were similar compared to the results generated using PLINK (**Supplementary Data 18**).”

Minor points:

4. (Abstract) In line with the first point, my suggestion would be to mention the number of identified loci of the meta-analysis including IHGC, and the number of novel loci of this combined GWAS compared to Gormley et al.

As suggested by the reviewer, we have now revised the abstract as follows:

“To elucidate the etiology of this common disorder, we conducted a multiethnic genome-wide association meta-analysis of migraine, combining results from the GERA and UK Biobank cohorts, followed by a European-ancestry meta-analysis using public summary statistics. We report 79 loci associated with migraine, of which 45 were novel.”

5. (Introduction) ‘[...] and severe disruptions of the brain parenchyma’.

It is not generally accepted that the brain parenchyma is severely disrupted by migraine, the authors could consider to rephrase this sentence.

We have now rephrased this sentence in the introduction as follows:

“Migraine is a common disabling disorder ~~characterized by episodic acute and severe disruptions of the brain parenchyma~~ that can be accompanied by a wide range of symptoms of varying intensity, including headache pain that is often one-sided, and accompanied by nausea, sound and light sensitivity, and disturbed vision.”

6. (Results – Multiethnic meta-analysis of GERA and UKB) Similar to the replication section, it would be interesting to replicate the UKB+GERA findings in the IHGC GWAS using a Bonferroni significance threshold.

This is an excellent suggestion. However, as the GWAS summary statistics data, which are publicly accessible, report only lead SNPs from the Gormley et al. study with a P-value of less than 1.0×10^{-5} , some of the strongest SNPs reported by Gormley et al. were different than ours. Further, some of our SNPs within the novel loci identified may replicate at a Bonferroni significance threshold in IHGC ($P < 0.05/10$ novel loci = 5.0×10^{-3}), however, they were not reported in the publicly accessible GWAS summary statistic from the Gormley study because of the significance threshold ($P < 1.0 \times 10^{-5}$).

We have now reported a replication of the GERA+UKB in the IHGC GWAS of migraine from the study of Gormley et al. (Nature Genetics 2016). Out of the 10 novel loci that we identified in the combined (GERA+UKB) GWAS multiethnic meta-analysis, 6 were available in the IHGC GWAS summary statistics, even though the strongest SNPs reported by Gormley et al. were different than ours. We have now added some text in the Results section as below and added a new Supplementary Data (**Supplementary Data 2**) to report those replication results.

“Replication in the IHGC data

We then tested the ten lead SNPs representing each of the ten novel loci for replication in the most recent large genetic study of migraine conducted by the IHGC⁵. However, as the GWAS summary statistics data, publicly accessible, reported only the lead SNPs from the Gormley et al. study with a P-value of less than 1.0×10^{-5} , some of the strongest SNPs reported by Gormley et al. were different than ours. Six loci, including *TMEM51*, *MIR4791-EFHB*, *LINC00472-RIMS1*, *FXN*, *GATA3-SFTA1P*, and *LINC00310-KCNE2*, replicated at Bonferroni significance ($P < 0.05/10$ novel loci = 5.0×10^{-3}) (**Supplementary Data 2**). Our lead SNPs within the remaining 4 novel loci (i.e. *SLC45A1/RERE*, *MARGPRE-ZNF195*, *CALCB*, and *B3GNTL1-METRNL*) were not reported in the publicly accessible GWAS summary statistic from the Gormley et al. study, however those may replicate at a Bonferroni significance threshold in IHGC ($1.0 \times 10^{-5} \leq P < 5.0 \times 10^{-3}$) but were not publicly accessible.”

7. (Results – Replication of previous migraine GWAS results) My suggestion would be to only to consider the meta-analysis performed by Gormley et al. since this is the most recent and most complete GWAS which includes the previously performed GWAS.

As suggested by the reviewer, we now consider for replication only 38 lead SNPs reported in the most recent and exhaustive GWAS of migraine conducted to date (Gormley et al. Nature Genetics 2016). We have updated the Supplementary Data 3 and the text in the Results section as below:

“Replication of previous migraine GWAS results

We also investigated the lead SNPs within 38 loci associated with migraine at a genome-wide significance level from the most recent and exhaustive GWAS of migraine conducted to date⁵ (Supplementary Data 3). Ten lead SNPs of the 36 available replicated at a genome-wide level of significance in our combined (GERA+UKB) multiethnic meta-analysis (including rs10218452 at *PRDM16*, rs2078371 near *TSPAN2/NGF*, rs1925950 at *MEF2D*, rs10166942 at *TRPM8/HJURP*, rs9349379 at *PHACTR1*, rs28455731 near *GJA1*, rs186166891 at *C7orf10*, rs6478241 at *ASTN2*, rs1024905 near *FGF6*, and rs11172113 at *LRP1*) (Supplementary Data 3). Further, 14 additional SNPs replicated at Bonferroni significance ($P < 0.05/36 = 1.39 \times 10^{-3}$), and 4 showed nominal evidence of association ($P < 0.05$). In contrast, 8 SNPs (including rs140002913 near *NOTCH4*, rs10155855 near *DOCK4/IMMP2L*, rs2506142 at *NRP1*, rs561561 at *IGSF9B*, rs75213074 near *WSCD1/NLRP1*, rs17857135 at *RNF213*, rs144017103 near *CCM2L/HCK*, and rs12845494 near *MED14/USP9X*) were not validated in the current combined (GERA+UKB) multiethnic meta-analysis ($P > 0.05$).

8. (Results – Ethnic-specific and conditional analyses) Please add the number of cases and controls for each ancestry-specific meta-analysis, as this is probably the most important reason why no additional loci have been identified in the non-European ancestry analyses.

As suggested by the reviewer, we now provide the number of cases and controls for each ancestry-specific meta-analysis in the Results section, as follows:

“Ethnic-specific and conditional analyses

For ethnic groups represented in each cohort, we conducted ethnic-specific meta-analyses of each group. In the European ancestry (GERA non-Hispanic whites + UKB Europeans + IHGC Europeans only; 85,726 migraine cases and 803,292 controls) meta-analysis, we identified 73 loci, (...). Conducting a GWAS meta-analysis of East Asian-specific cohorts (GERA+UKB East Asian ancestry individuals only; 569 migraine cases and 6,619 controls) and a GWAS meta-analysis of African-specific cohorts (GERA+UKB African ancestry individuals only; 504 migraine cases and 10,104 controls) did not result in the identification of genome-wide significant findings.”

9. (Results – Ethnic-specific and conditional analyses) Although no novel loci were identified in the non-European ancestry analysis, could the authors provide some information on whether the identified loci in the multi-ethnic analysis was still present in the ancestry-specific analyses?

This is a good point raised by the reviewer. To clarify, no genome-significant loci were observed in the non-European ancestry meta-analyses (East Asian-specific or African-specific cohorts). We have modified the text in the Results section and added a sentence to reflect the most likely limited power in these analyses to detect significant effects:

“Ethnic-specific and conditional analyses

For ethnic groups represented in each cohort, we conducted ethnic-specific meta-analyses of each group. In the European ancestry (GERA non-Hispanic whites + UKB Europeans + IHGC Europeans only; 85,726 migraine cases and 803,292 controls) meta-analysis, we identified 73 loci, of which 35 were additional novel (Supplementary Data 4). To identify independent signals within the 73 genomic regions identified in the European-specific meta-analysis, we performed a multi-SNP-based conditional & joint association analysis (COJO)³¹, which revealed 2 additional independent SNPs within the known loci *TSPAN2-NGF* (rs2207237) and *ADAMTSL4-ECM1* (rs7524797). Conducting a GWAS meta-analysis of East Asian-specific cohorts (GERA+UKB East Asian ancestry individuals only; 569 migraine cases and 6,619 controls) and a GWAS meta-analysis of African-specific cohorts (GERA+UKB African ancestry individuals only; 504 migraine cases and 10,104 controls) did not result in the identification of ~~additional novel~~ genome-wide significant findings. We may have been underpowered to detect effects with statistical significance in those non-European ancestry meta-analyses.”

10. (Results – SNP prioritization and annotations) Could the authors provide some more more information on the two likely causal variants, e.g. are they exonic/intronic, what is their CADD score, etc.?

As suggested by the reviewer, we have checked this “functional consequence” information on NCBI dbSNP or Genome Data Viewer for the five potential causal variants (with >95% probability) and we have now specified this information in the Results section as follows:

“SNP prioritization and annotations

To prioritize variants within the 22 loci identified in the combined multiethnic (GERA+UKB) meta-analysis and within the 73 loci identified in the European-specific meta-analysis, we applied a Bayesian approach (CAVIARBF)²⁹. (...). For the 22 loci identified in the combined multiethnic (GERA+UKB) meta-analysis, two sets included a unique variant (**Supplementary Data 8**). These include the previously reported intronic variant rs9349379 at *PHACTR1*⁵, and the newly identified intergenic variant rs13087932 at *MIR4791-EFHB* with 100.0% and 97.2% posterior probability of being the causal variants, respectively, suggesting that these variants are more likely to be the true causal variants. For the 73 loci identified in the European-specific meta-analysis, four sets included a unique variant (**Supplementary Data 9**). In addition to rs9349379 at *PHACTR1*, we found that the intronic variants rs5763529 at *ASCC2* and rs11172113 at *LRP1*, and the intergenic variant rs28451064 at *LINC00310-KCNE2* were more likely to be the true causal variants with 100.0%, 99.9%, and 97.3% posterior probability, respectively.”

As suggested by the reviewer, we have also looked for the Combined Annotation Dependent Depletion (CADD) score for those five potential causal variants and reported it in the table below:

Locus	SNP	Functional consequence	CADD Score
PHACTR1	rs9349379	intronic	5.090
MIR4791-EFHB	rs13087932	intergenic	1.505
ASCC2	rs5763529	intronic	4.419
LRP1	rs11172113	intronic	12.88
LINC00310-KCNE2	rs28451064	intergenic	14.20

11. (Results – Gene-Based Association Analysis) Since gene-based analyses take a different approach and have a more lenient threshold, were there also genes identified outside the originally identified loci?

This is a good point raised by the reviewer. We have checked whether some of the 47 significant genes ($P < 2.51 \times 10^{-6}$ (0.05/19,933)) that we prioritized based on our gene-based association analysis were outside the loci identified in the current study (i.e. originally 22 loci from the multiethnic (GERA+UKB) meta-analysis, 73 loci from the European ancestry meta-analysis (GERA+UKB+IHGC), or 3 loci from the female-specific analysis). Nine of those 47 genes were outside the loci identified in the current study. We have added a sentence in the Results section to reflect this point:

“Gene-Based Association Analysis and Gene Prioritization

To identify additional genes associated with migraine at a gene level, we conducted a gene-based association analysis using the functional mapping and annotation of genetic associations (FUMA)³² integrative tool (...). We found significant associations with migraine for 47 genes, with the strongest association for *STAT6* ($P = 1.24 \times 10^{-23}$), followed by *UFL1* ($P = 7.26 \times 10^{-19}$), and *FHL5* ($P = 1.25 \times 10^{-18}$) (**Supplementary Data 10**). Out of the 47 genes, 9 were located outside the loci identified in the current study, including *PRKCE*, *RCHY1*, *THAP6*, *MAPK9*, *RP11-508N12.4*, *LRCH1*, *PNKP*, *AKT1S1*, and *TBC1D17*.”

12. (Results – Genetic correlation between migraine and other phenotypes) Why was a P-value threshold of $< 5 \times 10^{-8}$ used for this analysis?

We thank the reviewer for catching this typo in the Results section. Genetic correlations were considered significant after Bonferroni adjustment for multiple testing ($P < 6.48 \times 10^{-5}$ which corresponds to 0.05/772 phenotypes tested). We have now corrected the P-value in the Results section as follows:

“Genetic correlation between migraine and other phenotypes

Genome-wide genetic correlations of migraine were calculated with a total of 772 complex traits and diseases (...). A total of 75 significant genetic correlations were observed ($P < 6.48 \times 10^{-5}$ which corresponds to 0.05/772 phenotypes

tested; Supplementary Data 16). Among those 75 genetic correlations, 38 reached genome-wide level of significance (Supplementary Figure 5).”

13. (Discussion) Do the authors think the genetic correlation with neck, shoulder and back pain may in part be explained by a misclassification of migraine cases, with some being actually tension headaches rather than migraine?

This is a good point raised by the reviewer. Although migraine cases in GERA were identified in the KPNC electronic health record system using our previously described and validated migraine probability algorithm³³, which is based on migraine-specific prescriptions and International Classification of Disease, Ninth (ICD9) or Tenth Revision (ICD10) diagnosis codes, most of the migraine cases in UKB were based on self-reported data. This may lead to phenotype misclassification in UKB. So, it is possible that misclassification of the migraines cases occurred in UKB, and consequently affected the genetic correlation results. We have added some text in the Discussion to reflect this point as follows:

“We recognize several potential limitations of our study. First, it is important to note phenotypic differences for migraine between the 2 study cohorts. Although migraine cases in GERA were identified in the KPNC electronic health record system using our previously described and validated migraine probability algorithm³³, which is based on migraine-specific prescriptions and International Classification of Disease, Ninth (ICD9) or Tenth Revision (ICD10) diagnosis codes, most of the migraine cases in UKB were based on self-reported data. This may lead to phenotype misclassification which may have affected, for instance, the high positive genetic correlation between migraine and neck, shoulder or back pain.”

References

1. Anttila V, Stefansson H, Kallela M, Todt U, Terwindt GM, Calafato MS, Nyholt DR, Dimas AS, Freilinger T, Muller-Myhsok B, et al. Genome-wide association study of migraine implicates a common susceptibility variant on 8q22.1. *Nat Genet.* 2010;42:869-73.
2. Anttila V, Winsvold BS, Gormley P, Kurth T, Bettella F, McMahon G, Kallela M, Malik R, de Vries B, Terwindt G, et al. Genome-wide meta-analysis identifies new susceptibility loci for migraine. *Nat Genet.* 2013;45:912-917.
3. Chasman DI, Schurks M, Anttila V, de Vries B, Schminke U, Launer LJ, Terwindt GM, van den Maagdenberg AM, Fendrich K, Volzke H, et al. Genome-wide association study reveals three susceptibility loci for common migraine in the general population. *Nat Genet.* 2011;43:695-8.
4. Freilinger T, Anttila V, de Vries B, Malik R, Kallela M, Terwindt GM, Pozo-Rosich P, Winsvold B, Nyholt DR, van Oosterhout WP, et al. Genome-wide association analysis identifies susceptibility loci for migraine without aura. *Nat Genet.* 2012;44:777-82.
5. Gormley P, Anttila V, Winsvold BS, Palta P, Esko T, Pers TH, Farh KH, Cuenca-Leon E, Muona M, Furlotte NA, et al. Meta-analysis of 375,000 individuals identifies 38 susceptibility loci for migraine. *Nat Genet.* 2016;48:856-66.
6. Chang X, Pellegrino R, Garifallou J, March M, Snyder J, Mentch F, Li J, Hou C, Liu Y, Sleiman PMA, et al. Common variants at 5q33.1 predispose to migraine in African-American children. *J Med Genet.* 2018;55:831-836.
7. Nyholt DR, International Headache Genetics C, Anttila V, Winsvold BS, Kurth T, Stefansson H, Kallela M, Malik R, Vries B, Terwindt GM, et al. Concordance of genetic risk across migraine subgroups: Impact on current and future genetic association studies. *Cephalalgia.* 2015;35:489-99.
8. Pers TH, Karjalainen JM, Chan Y, Westra HJ, Wood AR, Yang J, Lui JC, Vedantam S, Gustafsson S, Esko T, et al. Biological interpretation of genome-wide association studies using predicted gene functions. *Nat Commun.* 2015;6:5890.
9. Sauvageau G, Lansdorp PM, Eaves CJ, Hogge DE, Dragowska WH, Reid DS, Largman C, Lawrence HJ, Humphries RK. Differential expression of homeobox genes in functionally distinct CD34+ subpopulations of human bone marrow cells. *Proc Natl Acad Sci U S A.* 1994;91:12223-7.
10. Manzanares M, Nardelli J, Gilardi-Hebenstreit P, Marshall H, Giudicelli F, Martinez-Pastor MT, Krumlauf R, Charnay P. Krox20 and kreisler co-operate in the transcriptional control of segmental expression of Hoxb3 in the developing hindbrain. *EMBO J.* 2002;21:365-76.
11. Chan KK, Chen YS, Yau TO, Fu M, Lui VC, Tam PK, Sham MH. Hoxb3 vagal neural crest-specific enhancer element for controlling enteric nervous system development. *Dev Dyn.* 2005;233:473-83.
12. Sham MH, Vesque C, Nonchev S, Marshall H, Frain M, Gupta RD, Whiting J, Wilkinson D, Charnay P, Krumlauf R. The zinc finger gene Krox20 regulates HoxB2 (Hox2.8) during hindbrain segmentation. *Cell.* 1993;72:183-96.
13. Komuves LG, Shen WF, Kwong A, Stelnicki E, Rozenfeld S, Oda Y, Blink A, Krishnan K, Lau B, Mauro T, et al. Changes in HOXB6 homeodomain protein structure and localization during human epidermal development and differentiation. *Dev Dyn.* 2000;218:636-47.

14. Hromatka BS, Tung JY, Kiefer AK, Do CB, Hinds DA, Eriksson N. Genetic variants associated with motion sickness point to roles for inner ear development, neurological processes and glucose homeostasis. *Hum Mol Genet.* 2015;24:2700-8.
15. Ishizaki K, Takeshima T, Fukuhara Y, Araki H, Nakaso K, Kusumi M, Nakashima K. Increased plasma transforming growth factor-beta1 in migraine. *Headache.* 2005;45:1224-8.
16. Saygi S, Alehan F, Erol I, Yalcin YY, Atac FB, Kubat G. TGF-beta1 genotype in pediatric migraine patients. *J Child Neurol.* 2015;30:27-31.
17. Varnum-Finney B, Purton LE, Yu M, Brashem-Stein C, Flowers D, Staats S, Moore KA, Le Roux I, Mann R, Gray G, et al. The Notch ligand, Jagged-1, influences the development of primitive hematopoietic precursor cells. *Blood.* 1998;91:4084-91.
18. Walker L, Lynch M, Silverman S, Fraser J, Boulter J, Weinmaster G, Gasson JC. The Notch/Jagged pathway inhibits proliferation of human hematopoietic progenitors in vitro. *Stem Cells.* 1999;17:162-71.
19. Karanu FN, Murdoch B, Gallacher L, Wu DM, Koremoto M, Sakano S, Bhatia M. The notch ligand jagged-1 represents a novel growth factor of human hematopoietic stem cells. *J Exp Med.* 2000;192:1365-72.
20. Elliott GC, Gurtu R, McCollum C, Newman WG, Wang T. Foramen ovale closure is a process of endothelial-to-mesenchymal transition leading to fibrosis. *PLoS One.* 2014;9:e107175.
21. Sztajzel R, Genoud D, Roth S, Mermillod B, Le Floch-Rohr J. Patent foramen ovale, a possible cause of symptomatic migraine: a study of 74 patients with acute ischemic stroke. *Cerebrovasc Dis.* 2002;13:102-6.
22. van Rossum D, Hanisch UK, Quirion R. Neuroanatomical localization, pharmacological characterization and functions of CGRP, related peptides and their receptors. *Neurosci Biobehav Rev.* 1997;21:649-78.
23. Russo AF. Calcitonin gene-related peptide (CGRP): a new target for migraine. *Annu Rev Pharmacol Toxicol.* 2015;55:533-52.
24. Pellesi L, Guerzoni S, Pini LA. Spotlight on Anti-CGRP Monoclonal Antibodies in Migraine: The Clinical Evidence to Date. *Clin Pharmacol Drug Dev.* 2017;6:534-547.
25. Edvinsson L. CGRP Antibodies as Prophylaxis in Migraine. *Cell.* 2018;175:1719.
26. Edvinsson L, Haanes KA, Warfvinge K, Krause DN. CGRP as the target of new migraine therapies - successful translation from bench to clinic. *Nat Rev Neurol.* 2018;14:338-350.
27. King CT, Gegg CV, Hu SN, Sen Lu H, Chan BM, Berry KA, Brankow DW, Boone TJ, Kezunovic N, Kelley MR, et al. Discovery of the Migraine Prevention Therapeutic Aimovig (Erenumab), the First FDA-Approved Antibody against a G-Protein-Coupled Receptor. *ACS Pharmacol Transl Sci.* 2019;2:485-490.
28. Tepper SJ. History and Review of anti-Calcitonin Gene-Related Peptide (CGRP) Therapies: From Translational Research to Treatment. *Headache.* 2018;58 Suppl 3:238-275.
29. Chen W, Larrabee BR, Ovsyannikova IG, Kennedy RB, Haralambieva IH, Poland GA, Schaid DJ. Fine Mapping Causal Variants with an Approximate Bayesian Method Using Marginal Test Statistics. *Genetics.* 2015;200:719-36.
30. Mbatchou J, Barnard L, Backman J, Marcketta A, Kosmicki JA, Ziyatdinov A, Benner C, O'Dushlaine C, Barber M, Boutkov B, et al. Computationally efficient whole genome regression for quantitative and binary traits. *bioRxiv.* 2020:2020.06.19.162354.
31. Yang J, Ferreira T, Morris AP, Medland SE, Genetic Investigation of ATC, Replication DIG, Meta-analysis C, Madden PA, Heath AC, Martin NG, et al. Conditional and joint multiple-SNP analysis of GWAS summary statistics identifies additional variants influencing complex traits. *Nat Genet.* 2012;44:369-75, S1-3.
32. Watanabe K, Taskesen E, van Bochoven A, Posthuma D. Functional mapping and annotation of genetic associations with FUMA. *Nat Commun.* 2017;8:1826.
33. Pressman A, Jacobson A, Eguilos R, Gelfand A, Huynh C, Hamilton L, Avins A, Bakshi N, Merikangas K. Prevalence of migraine in a diverse community--electronic methods for migraine ascertainment in a large integrated health plan. *Cephalalgia.* 2016;36:325-34.

REVIEWERS' COMMENTS:

Reviewer #1 (Remarks to the Author):

I thank the Authors for the careful and considerable efforts to address my comments.

I am satisfied with the additional analyses and amendments and have only a few minor requests:

1) When the authors discuss their comparison of PLINK with REGENIE results - i.e., line 376-377: "The GWAS results generated using REGENIE were similar compared to the results generated using PLINK (Supplementary Data 18)"; can they please confirm and note that the overall meta-analysis results, in particular, the genome-wide significant loci, remain (the same).

2) When the authors describe and discuss the IHGC summary statistics - e.g., line 66-68: "However, as the GWAS summary statistics data, publicly accessible, reported only the lead SNPs from the Gormley et al. study with a P-value of less than 1.0×10^{-5} "; can they please not use "lead" here (and elsewhere when referring to these data), because the IHGC summary statistics file contain all SNPs with P-value of less than 1.0×10^{-5} . The use of "lead" implies they are the "index" or "LD-independent" SNPs at the loci.

3) Related to the p-value threshold issue raised by Reviewer #3, the Authors also need to amend the phrasing around the genetic correlation significance, including the Supplementary Figure 5 legend (i.e., "Only entries that reached a genome-level of significance are shown in this figure") to be in line with the main text. That is, rather than "genome-wide level of significance", the Authors should state "study-wide level of significance" or "experiment-wide level of significance" when referring to whether the genetic correction result is significant ($p < 0.05$) after adjusting for the total number of genetic correlation tests performed.

4) It might be preferable to report allele frequencies to 4 decimal places when comparing the male and female EAF and amend the phrasing. Specifically, "We found that the EAF of each SNP were near identical across the 4 sub-groups" is not what you want to say, because if this was strictly true, the SNPs would not be associated with migraine!

Reviewer #3 (Remarks to the Author):

I would like to thank the authors for their helpful responses; I have no additional comments or suggestions.